# GeoPixel🤖: Pixel Grounding Large Multimodal Model in Remote Sensing

**Akashah Shabbir** [1]  **Mohammed Zumri** [1]  **Mohammed Bennamoun** [2]  **Fahad Shahbaz Khan** [1 3]  **Salman Khan** [1 4]

## Abstract

*Recent advances in large multimodal models (LMMs) have recognized fine-grained grounding as an imperative factor of visual understanding and dialogue. However, the benefits of such representation in LMMs are limited to the natural image domain, and these models perform poorly for remote sensing (RS). The distinct overhead viewpoint, scale variation, and presence of small objects in high-resolution RS imagery present a unique challenge in region-level comprehension. Moreover, the development of the grounding conversation capability of LMMs within RS is hindered by the lack of granular, RS domain-specific grounded data. Addressing these limitations, we propose GeoPixel - the first end-to-end high-resolution RS-LMM that supports pixel-level grounding. This capability allows fine-grained visual perception by generating interleaved masks in conversation. GeoPixel supports up to 4K HD resolution in any aspect ratio, ideal for high-precision RS image analysis. To support the grounded conversation generation (GCG) in RS imagery, we curate a visually grounded dataset GeoPixelD through a semi-automated pipeline that utilizes set-of-marks prompting and spatial priors tailored for RS data to methodically control the data generation process. GeoPixel demonstrates superior performance in pixel-level comprehension, surpassing existing LMMs in both single-target and multi-target segmentation tasks. Our methodological ablation studies validate the effectiveness of each component in the overall architecture. https://github.com/mbzuai-oryx/GeoPixel.*

---

[1]Mohamed bin Zayed University of Artificial Intelligence [2]The University of Western Australia [3]Linköping University [4]Australian National University. Correspondence to: Akashah Shabbir <akashah.shabbir@mbzuai.ac.ae>, Mohammed Zumri <mohammed.zumri@mbzuai.ac.ae>.

*Proceedings of the 42nd International Conference on Machine Learning*, Vancouver, Canada. PMLR 267, 2025. Copyright 2025 by the author(s).

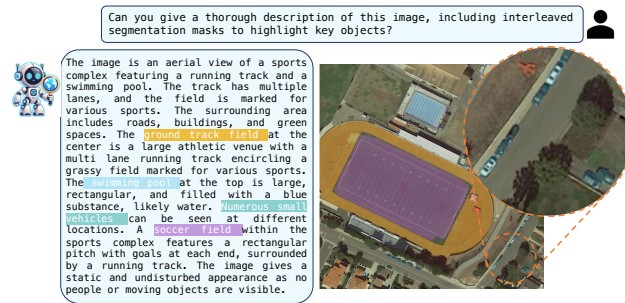

*Figure 1.* An example of visually grounded detailed descriptions generated by the proposed GeoPixel, highlighting its ability to interpret and segment high-resolution remote sensing imagery with fine-grained precision. The model applies distinct masks to key objects (ground track field, swimming pool, soccer field) and semantic mask to smaller objects (vehicles). It effectively identifies spatial positions (center, top) and relationships (within the sports complex) while distinguishing between the global context (buildings, roads, green spaces) and localized structures.

## 1. Introduction

Recent large multimodal models (LMMs) (Liu et al., 2024a; Dai et al., 2023; Bai et al., 2023b; Chen et al., 2024b) have utilized the foundational capabilities of Large Language Models (LLMs) (Touvron et al., 2023; Chiang et al., 2023; Javaheripi et al.; Bai et al., 2023a) and successfully expanded their horizon to the visual modality with promising capabilities. These LMMs can not only perform visual recognition, but also excel in advanced perception and reasoning required for vision-language tasks such as visual question answers, image captioning, visual grounding, and referring expression segmentation. Grounding LMMs (Rasheed et al., 2024; Ma et al., 2025; Zhao et al., 2023) have further advanced the fine-grained context-aware interpretation of complex visual information by allowing textual outputs to be associated with object instances. Facilitated by large-scale data in the natural images domain, grounding multimodal models pre-trained on extensive datasets have shown impressive capabilities, achieving performance levels comparable to specialist models.

However, with increasing granularity of vision and language understanding, these general domain models exhibit significant limitations in adequately supporting complex earth observation tasks. The performance degradation is influ-

*Table 1.* Comparison of remote sensing large multimodal models (RS-LMMs), focusing on their grounding capabilities. The 'Region Output' column highlights the model's ability to associate objects with specific spatial regions. Existing models primarily utilize LLMs to generate bounding box coordinates for object grounding. However, none of the current RS-LMMs possess the capability for 'pixel grounding', i.e., generating detailed segmentation masks, which are crucial for fine-grained spatial interpretation.

| MODELS | RESOLUTION | IMAGE | REGION OUTPUT | REGION DECODER | PIXEL GROUNDING |
|---|---|---|---|---|---|
| RSGPT (HU ET AL., 2023) | 224 × 224 | ✓ | ✗ | ✗ | ✗ |
| H2RSVLM (PANG ET AL., 2024) | 336 × 336 | ✓ | ✗ | ✗ | ✗ |
| RS-LLAVA (BAZI ET AL., 2024) | 336 × 336 | ✓ | ✗ | ✗ | ✗ |
| GEOCHAT (KUCKREJA ET AL., 2024) | 504 × 504 | ✓ | ✓ | ✗ | ✗ |
| SKYEYEGPT (ZHAN ET AL., 2024) | 448 × 448 | ✓ | ✓ | ✗ | ✗ |
| EARTHGPT (ZHANG ET AL., 2024C) | - | ✓ | ✓ | ✗ | ✗ |
| LHRS-BOT (MUHTAR ET AL., 2024) | 224×224 | ✓ | ✓ | ✗ | ✗ |
| SKYSENSEGPT (LUO ET AL., 2024) | 504 × 504 | ✓ | ✓ | ✗ | ✗ |
| GEOPIXEL | DYNAMIC UPTO 4K | ✓ | ✓ | ✓ | ✓ |

enced not only by the unique vantage point inherent to remote sensing (RS) images but also by large variations in the objects' size and orientation. Moreover, in high-resolution (HR) remote sensing imagery, objects of interest may exhibit challenging-to-segment spatial footprints, such as narrow bridges that connect urban landscapes and play a critical role in city traffic planning, adding further complexity to the task.

Existing vision language models in RS (Luo et al., 2024; Zhang et al., 2024b; Kuckreja et al., 2024) use quantized coordinates in the form of bounding boxes to localize and ground objects in their response. Such a representation structure is not adequate to associate correct object semantics and also adds a computational burden to the LLM that scales with the number of distinguishable objects. Moreover, monitoring the geospatial environment and its entities demands a broader spatial perspective, now increasingly achievable through advancements in RS technologies that provide HR imagery. However, despite the availability of such rich data, current LMMs in RS struggle to fully exploit this spatial detail. These models often struggle with suboptimal resolution capabilities, hindering their ability to capture the intricate patterns present in high-resolution RS images. In addition, existing RS datasets often lack fine-grained spatial association between objects and their corresponding linguistic descriptions.

To address these issues, we present GeoPixel, a model that can generate a detailed natural language response for a high-resolution RS image with corresponding geospatial object segmentation masks. Our contributions are as follows:

- Our proposed LMM, GeoPixel, is explicitly designed for high-resolution RS image analysis with advanced multi-target pixel grounding capability. Our model adaptively divides the input images into local and global regions, enabling efficient encoding and analysis

by accommodating up to 4k resolution.

- We create GeoPixelD, a multi-modal grounded conversation generation (GCG) dataset comprising 53,816 grounded phrases linked to 600,817 object masks, specifically tailored for RS image understanding. Extensively granular annotations are created with segmentation masks through a semi-automated, scalable pipeline that integrates prior-informed visual prompting with state-of-the-art LMMs and ensures quality via rigorous verification and filtering steps.

- We introduce a comprehensive benchmark for evaluating RS LMMs in fine-grained visual understanding, assessing models' ability to interpret complex, spatially grounded information. It comprises 5,427 manually validated referring expressions and segmentation masks, covering 61,384 annotated objects.

## 2. Related Work

**Large Multimodal Models (LMMs):** LMMs build on the success of LLMs to acquire vision capabilities. Pioneer works such as LLaVA (Liu et al., 2024b), MiniGPT-4 (Zhu et al., 2023), InstructBLIP (Dai et al., 2023) and mPLUG-Owl (Ye et al., 2023b) aligned visual features with language representations through a vision language connector, enhanced by instruction tuning to improve multimodal integration. Improving beyond image-level understanding, models such as GPT4RoI (Zhang et al., 2023), InternGPT (Liu et al., 2023b) and RegionGPT (Guo et al., 2024) introduce regional understanding by allowing inputs such as points, masks, and bounding boxes. Some models feed image coordinates directly into the language model, while others employ additional feature extraction modules to represent specific image regions' features effectively.

**Grounding LMMs:** Region-level comprehension is further expanded by models such as Kosmos-2 (Peng et al., 2024), Ferret (You et al., 2023), Shikra (Chen et al., 2023), Pink (Xuan et al., 2024) and LION (Chen et al., 2024a) that allow for the precise location of objects in their outputs based on textual descriptions, a capability known as grounding. These models localize objects on a coarse scale using bounding boxes. Recent models (Lai et al., 2024; Rasheed et al., 2024; Xia et al., 2024; Ren et al., 2024; Zhang et al., 2024d; Liu et al., 2023a) focus on achieving more fine-grained visual and linguistic semantic alignment, by exploring pixel grounding. LISA (Lai et al., 2024), PixelLM (Ren et al., 2024) and GLaMM (Rasheed et al., 2024) incorporate a [SEG] token into the LLM's vocabulary, leveraging its corresponding token embedding as a conditioning input for SAM (Kirillov et al., 2023) to enable segmentation. Additionally, GSVA (Xia et al., 2024) introduces a [REJ] token to explicitly learn to reject specified targets. Whereas

Llava-plus (Liu et al., 2023a) employs LLMs as agents to assign tasks to the segmentation expert.

Our work aligns with pixel-grounding approaches, such as those in (Lai et al., 2024; Ren et al., 2024; Rasheed et al., 2024). However, these models do not interpret the distinct top-down perspective and cannot differentiate complex spatial arrangements of remote sensing (RS) imagery. In addition, the models' restricted input size, typically limited to dimensions such as 224×224, exacerbates this issue by constraining the field of view and spatial perception.

**High-Resolution Understanding:** Vision encoders, such as CLIP ViT (Radford et al., 2021), are widely utilized for various vision tasks but are typically constrained by low resolution (e.g. 224×224) restricting their applicability in HR scenarios. To address this limitation, some approaches (Dosovitskiy et al., 2021; Bai et al., 2023b; Li et al., 2023) scale positional encodings within the CLIP model through interpolation to accommodate larger input sizes, while others such as CogAgent (Hong et al., 2024) and Vary (Wei et al., 2025), employ an additional HR branch. Models such as Monkey (Li et al., 2024b), SPHNIX (Lin et al., 2023), Llava-Next (Liu et al., 2024a), IXC2.5 (Zhang et al., 2024a), Textmonkey (Liu et al., 2024d) and Ureader (Ye et al., 2023a) divide the image into grids to enhance performance on HR text-centric tasks.

**Remote Sensing (RS) LMMs:** RSGPT (Hu et al., 2023) pioneered RS-based natural language conversation and generated detailed captions, followed by GeoChat (Kuckreja et al., 2024) which supported region-specific inputs and visual grounding through oriented bounding box coordinates. SkyEyeGPT (Zhan et al., 2024) extended its functionality to RS video captioning, while EarthGPT (Zhang et al., 2024c) and EarthDial (Soni et al., 2024) integrated multisensor RS interpretation. RS-LLaVA (Bazi et al., 2024) and H2RSVLM (Pang et al., 2024) further improved the RS data interpretation, with H2RSVLM uniquely recognizing and rejecting unanswerable questions. SkySenseGPT (Luo et al., 2024) enables image-level scene graph generation and relation reasoning, while LHRS-Bot (Muhtar et al., 2024) enhances multilevel vision-language alignment and TeoChat (Irvin et al., 2025) converses on temporal sequences. However, these models operate at low resolution and lack pixel-level understanding and grounding.

GeoGround (Zhou et al., 2024), RSUniVLM (Liu & Lian, 2024), and GeoPix (Ou et al., 2025) are works concurrent to ours and share similarities. GeoGround and RSUniVLM support pixel-level grounding by converting masks into text sequences, adding a computational burden to the LLM that scales with the number of distinguishable objects. Whereas, GeoPix incorporated both referring expressions segmentation and visual grounding/detection using box coordinates in text, which leads to task confusion.

## 3. Method

In the current remote sensing landscape, LMMs face significant limitations in terms of grounding and resolution capabilities (as seen in Table 1). Specifically, the outputs generated by these models lack precise spatial and semantic association with the imagery, leading to either ungrounded or only coarsely grounded text. Furthermore, most LMMs operate on relatively low-resolution data, which restricts their ability to perform fine-scale analysis essential for RS tasks such as detailed land use and transportation network extraction, infrastructure mapping, damage assessment, and environmental monitoring. To address these limitations, we present GeoPixel, a model designed to interpret high-resolution remote sensing images and generate finely detailed, pixel-grounded outputs that encompass multiple target objects.

### 3.1. GeoPixel Architecture Overview

GeoPixel primarily consists of 5 components (see Figure 2). (1) Adaptive Image Divider (2) Vision Encoder (3) Large Language Model (4) Grounding Vision Encoder (5) Pixel Decoder. The first three components are discussed in Section 3.2, while the latter two in Section 3.3. Jointly, these modules enable high-resolution perception, fine-grained interpretation, and grounding, as detailed below.

### 3.2. High Resolution Understanding

For high resolution, we adopt the dynamic image partitioning strategy of IXC-2.5 (Zhang et al., 2024a). Initially, the adaptive image divider processes the input image $x_{img}$, with dimensions $[h_i \times w_i]$, by up-scaling and padding it to align with the closest grid size denoted as $[g_h \times g_w]$.

$$g_h = k_1 \times \mathcal{B}, \quad g_w = k_2 \times \mathcal{B}, \quad (1)$$
$$\text{s.t.}, k_1, k_2 \in \mathbb{N}, \quad k_1 \times k_2 \leq \mathcal{P}$$

where $\mathcal{B}$ is the base resolution of the vision encoder and $\mathcal{P}$ is the number of maximum allowable image patches. Subsequently, the image is divided into $k_1 \times k_2$ non-overlapping patches $x_{p_{i,j}}$, where $p = 0, 1, 2, \ldots, (k_1 \times k_2 - 1)$, and $i, j$ denote the row and column indices of each patch in the grid.

We employ the scaled CLIP ViT-L/14 (Zhang et al., 2024a) as our vision encoder ($\mathcal{I}$), with a base resolution of $\mathcal{B} = 560$, facilitating large patches for enhanced visual representation. Furthermore, a global view $x_{glob}$ is generated by resizing $x_{img}$ to a fixed dimension of $560 \times 560$, aligned with the base resolution $\mathcal{B}$. Feature embeddings of patches $f_{p_{i,j}}$ are appended with a learnable token at the end of each row before flattening and merging (Dong et al., 2024b). Finally, global features $f_{glob}$ and patch features $f_p$ are concatenated ($||$) with a special separator ($s_g$) inserted between them

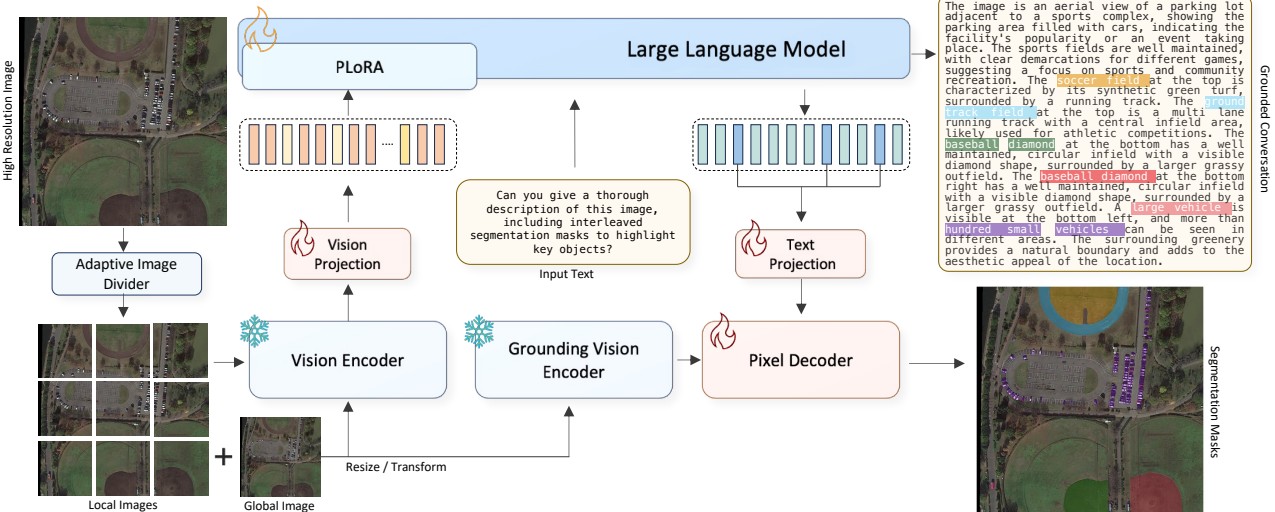

*Figure 2.* Overview of GeoPixel Architecture: Left: High-resolution RS images are dynamically partitioned into local patches and a resized global view, encoded by a frozen vision encoder. The encodings are projected into the language domain with separator tokens. Middle: Vision tokens, combined with text, are input into the LLM, where pLoRA is applied to vision tokens for efficient and effective multimodal alignment. Right: The corresponding embeddings for the [SEG] tokens are passed to a decoder through a text projector, along with vision embeddings from the grounding vision encoders, to generate precise segmentation masks.

(Ding et al., 2019), effectively integrating global semantics with fine-grained local details.

$$x_v = \mathcal{P}_v(f_{glob}||s_g||f_p) \qquad (2)$$

$$\text{s.t. } f_{glob} = \mathcal{I}(x_{glob}), f_{p_{i,j}} = \mathcal{I}(x_{p_{i,j}})$$

We project the final unified image features onto the LLM, InternLM2 7B model (Cai et al., 2024), denoted as $\mathcal{L}$, through a two-layer MLP as a vision projector $\mathcal{P}_v$. InternLM2 is an LLM designed to process sequences of text tokens, where its input consists of discrete embeddings derived from textual data. These embeddings correspond to either natural language tokens or special placeholders inserted to represent external modalities. The placeholder <IMAGE> in the input text query $x_t$ is a special token that represents the position of the image within the input sequence. When processing multimodal input, this placeholder is replaced with visual features $x_v$, extracted from the image, and projected into the same embedding space using $\mathcal{P}_v$.

Partial Low-Rank Adaptation (LoRA) (Dong et al., 2024a) is then applied to ensure efficient alignment of the vision tokens. Partial LoRA is a modality-specific plug-in module designed to align features from a new modality with LLM, preserving the model's inherent capabilities while enriching it with modality-specific insights. By applying low-rank adaptations selectively to visual tokens, Partial LoRA enhances alignment efficiency while reducing the computational cost. Formally, it introduces low-rank matrices $W_A \in \mathbb{R}^{C_r \times C_{in}}$ and $W_B \in \mathbb{R}^{C_{out} \times C_r}$ within each LLM linear layer, modifying the visual token outputs $x_v$

without altering the language token outputs $x_t$, thus achieving tailored cross-modal integration.

### 3.3. Pixel Grounding

To establish grounding in LMM, we initialize the grounding vision encoder ($\mathcal{I}_g$) with a pre-trained SAM-2 (Ravi et al., 2024) encoder together with a dedicated pixel decoder module ($\mathcal{D}$). The SAM2 visual encoder is a Masked Autoencoder (MAE) (He et al., 2022) pre-trained Hiera (Ryali et al., 2023) image encoder having a hierarchical structure that allows the use of multiscale features during decoding. The tokenizer's vocabulary is expanded by incorporating an additional <SEG> token, with its corresponding last-layer embedding ($E$) mapped to the decoder through a text projection layer $\mathcal{P}_t$. The text projection is a two-layer MLP that receives embeddings of dimension 4096 and transforms them into the input space of the pixel decoder, which has a dimensionality of 256.

The pixel decoder processes the image features from the frozen grounding vision encoder, along with projected LLM embeddings, to generate segmentation masks ($M$). The grounding vision encoder (SAM-2) is already pre-trained on large-scale datasets, making it highly effective at extracting robust, generalized image features for segmentation. Freezing the encoder ensures that these pretrained features are preserved. However, the light-weight pixel decoder and projection layer are trained to adapt pretrained vision features for segmentation tasks in GeoPixel.

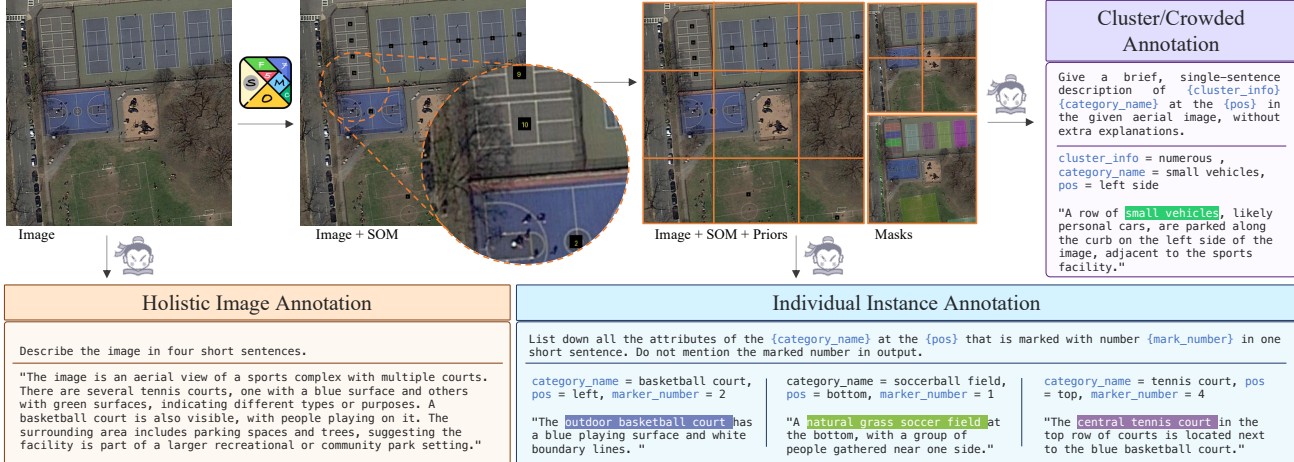

*Figure 3.* The GeoPixelD Annotation Pipeline provides detailed multi-tier descriptions of remote sensing imagery with object phrases aligned precisely with manually annotated masks. It begins with Holistic Image Annotation (bottom left), where an LMM generates concise scene descriptions. Individual Instance Annotation (bottom right) uses spatial({pos}) and categorical ({catagory_name}) priors with SOM ({mark_number}) prompting to describe key objects. Cluster Annotation (top right) organizes smaller or dense objects using refined grids for precise spatial analysis.

$$M = \mathcal{D}[\mathcal{I}_g(x_{img}), \mathcal{P}_t(E)] \qquad (3)$$

Given the variable length of the input image tokens, resulting from adaptive image partitioning, the output embedding mask for <SEG> tokens is dynamically adjusted to align with these variations. This configuration ensures accurate detection of the <SEG> token and its associated embedding.

## 4. GeoPixelD-RS Pixel Grounding Dataset

Remote sensing imagery captures intricate semantic information and complex inter-object relationships across diverse spatial scales. To enable LMMs to acquire a detailed comprehension ability, it is essential to integrate broad contextual views with object-level distinction. Addressing the current deficit in datasets capable of facilitating a fine-grained understanding of top-down perspectives, we introduce GeoPixelD, a dataset established to provide hierarchical descriptions derived through automated multilevel image analysis. GeoPixelD structures its descriptions at three primary levels: (1) holistic scene representation, (2) individual instance observations, and (3) densely populated object groups annotations (as depicted in Figure 3).

### 4.1. Holistic Image Annotation

Initially, we generated descriptive captions for RS images using a robust open source model, IXC (Zhang et al., 2024a), to capture comprehensive and diverse image details. We chose the IXC model (Zhang et al., 2024a) based on a comparative study conducted with other state-of-the-art vision language mod-

els, where IXC consistently outperformed its counterparts in terms of qualitative performance. Open-ended descriptions are restricted to a limited length, integrated in prompts like "<image> Describe the image in four short sentences" (Figure 3 (bottom left)). Thus, redundancy is effectively minimized in subsequent annotations, and the model is driven to provide a holistic, context-rich depiction of each image.

### 4.2. Individual Instance Annotation

Next, we identify key objects for depiction and employ set-of-mark (SOM) prompting (Yang et al., 2023), which overlays distinct visual markers over specific image regions, to provide auxiliary information for visually grounded output. However, directly employing this method for aerial imagery, characterized by expansive views and diverse objects within a single frame, poses challenges such as hallucinated markers and incorrectly associated details (see Figure 7). To enhance object description accuracy in complex RS images, we spatially guide the model by introducing prior knowledge in the query in the form of category name and location, along with a marked number. Thus, effectively guiding the model to generate a precise and comprehensive description.

Specifically, we partition each image into a 3×3 grid and localize the objects by measuring their overlap with the quadrants to determine their positional reference. This quadrant-based localization, combined with categorical labels and marked numbers, serves as positional and categorical priors for LMM, enhancing focus on the target objects, a process that proves effective given the densely packed and spatially complex nature of RS imagery, where objects often vary in

scale, orientation, and proximity.

We evaluated various open-source and proprietary models for prior-informed SOM prompting in RS imagery (see Figure 8), comparing combined and individual querying approaches. ChatGPT (OpenAI, 2023) generated detailed descriptions with inferred information, while Gemini (Team et al., 2023) and InternVL (Chen et al., 2024b) exhibited repetitive output as the target objects increased. InternLM-XComposer (Zhang et al., 2024a) achieved performance comparable to ChatGPT in terms of accuracy and diversity.

### 4.3. Cluster/Crowd Annotation

After identifying and annotating prominent large objects, the remaining objects are grouped or identified along with their spatial properties through a structured three-stage positional analysis. First, the image is divided into a 3×3 grid, with each grid cell assigned a unique identifier. To enhance alignment with human perceptual tendencies, the central region of the grid is given a larger spatial weight. Next, a 2×2 grid is considered for the localization of more dispersed objects. Finally, a half-image grid (1×2 and 2×1) assigns broader positional information. This gridding provides a systematic framework for the localization of clusters as well as large groups of objects. LMM then describes the group attributes using quantitative and positional information.

### 4.4. Unifying Annotations and Language Marking

For the preprocessed iSAID (Waqas Zamir et al., 2019) train dataset (Appendix A), we derive 16,795 holistic, 36,793 instance, and 17,023 group annotations, collectively encompassing 600,817 objects. The annotations were rigorously filtered to eliminate aerial perspective inconsistencies, artifacts such as marker identifiers, fore/background references, depth cues, and inconsistent descriptors.

The key noun chunk corresponding to the object category in individual- and group-level annotations is tagged with unique identifiers ('phrase-number'), each linked to an instance or semantic mask, a process termed *text marking*. These marked annotations are combined with holistic scene representations into a single descriptive narrative. We employ a Llama-3.1-instruct 8B (Dubey et al., 2024) LLM to paraphrase concatenated annotations while preserving their semantic integrity (see Figure 9). The LLM processes the concatenated text under strict constraints to retain all marked phrases unchanged, ensuring a consistent link to associated visual masks. The outputs undergo iterative paraphrasing if any marked phrases are not preserved. By adopting this language marking strategy, the GeoPixelD dataset achieves a robust framework to generate high-quality, context-rich GCG descriptions precisely aligned with visual elements.

Following similar procedures, test set GCG descriptions

(utilzing iSAID validation set images) undergoes meticulous manual curation, requiring ~350 man-hours to ensure annotation completeness. We employed a robust validation protocol where each gcg description in the test set was meticulously verified by expert annotators. The process includes correcting for any omissions, inaccuracies, or partial annotations, including adjustments to object attributes that do not align with the corresponding image, thereby establishing a high-quality evaluation benchmark.

## 5. Experiments

Here, we explain the implementation details, present a comparative performance analysis on Remote Sensing Grounded Conversation Generation (RS-GCG) and Referring Remote Sensing Image Segmentation (RRSIS), and include an ablation study to assess the impact of key components.

### 5.1. Implementation Details

The model is initialized with pre-trained InternLM-XComposer-2.5 model (IXC-2.5) with 7B parameters, using LoRA for efficient fine-tuning. A fixed CLIP ViT-L vision encoder with a resolution of 560×560 is employed, along with a grounded vision encoder initialized from SAM2 weights. The trainable components include a pixel decoder ($\mathcal{D}$), LoRA parameters ($\alpha = 8$), a vision projector $\mathcal{P}_v$, and a language projector $\mathcal{P}_t$. The adaptive image divider uses a maximum patch number $\mathcal{P}$ to 9 for training. In our training process, with an effective batch size of 20 over 10 epochs, the learning rate increases linearly to a maximum value of $3 \times 10^{-4}$ over the initial 100 training steps, followed by a gradual cosine decay. We train GeoPixel on the GeoPixelD dataset for the GCG task on two NVIDIA A6000-48GB GPUs, which takes around 3 days.

### 5.2. Baselines

To rigorously evaluate the efficacy of GeoPixel, we introduce three robust baselines for comparative analysis on the GeoPixelD benchmark. First baseline LISA†, an enhanced LISA (Lai et al., 2024) model, incorporates multitarget segmentation masks within its output pipeline and an updated tokenizer that includes phrase tokens (`<p>` and `</p>`) for the GCG task. The second baseline PixelLM†, derived from PixelLM (Ren et al., 2024), is configured without the SAM encoder, the codebook image feature scaling factor of 2, three segmentation tokens, and the vision tower resize parameter of 448. Phrase tokens are added, and `<SEG>` token is replaced with multiple codebook tokens. The third baseline, GLaMM, specifically GLaMM-GCG, a model tailored for the GCG task. LISA†, PixelLM† and GLaMM-ft models are initialized from pretrained LISA-7B-v1, PixelLM-7B and GLaMM-GCG (7B), respectively, and additionally trained on GeoPixelD data.

*Table 2.* Performance Comparison on RS-GCG task. LISA† and PixelLM† denote the pretrained LISA and PixelLM models adopted for RS-GCG and finetuned on GeoPixelD training data. GLaMM represents the zero-shot performance, whereas GLaMM-FT refers to the pretrained model finetuned on GeoPixelD. GeoPixel outperforms other models across all metrics.

| MODEL | CIDEr | METEOR | CLAIR | UNI-TARGET | | | MULTI-TARGET | | | OVERALL | | |
|---|---|---|---|---|---|---|---|---|---|---|---|---|
| | | | | AP50 | mIoU | RECALL | AP50 | mIoU | RECALL | AP50 | mIoU | RECALL |
| GLAMM (CVPR'24) | 0.1 | 5.8 | 43.11 | 1.2 | 18.1 | 14.8 | 0.5 | 16.5 | 6.3 | 0.5 | 16.9 | 7.1 |
| LISA† (CVPR'24) | 14.6 | 22.3 | 68.96 | 9.5 | 41.7 | 43.1 | 8.3 | 43.1 | 27.5 | 8.5 | 42.7 | 29.0 |
| PIXELLM† (CVPR'24) | 18.3 | 22.5 | 73.93 | 13.5 | 41.2 | 44.0 | 10.4 | 42.9 | 28.1 | 10.5 | 42.4 | 29.6 |
| GLAMM-FT (CVPR'24) | 15.7 | 23.0 | 71.74 | 18.8 | 44.4 | 48.5 | 12.4 | 47.1 | 31.1 | 12.5 | 46.4 | 32.8 |
| GEOPIXEL | **21.6** | **24.0** | **77.50** | **25.5** | **50.8** | **55.6** | **18.0** | **52.9** | **37.0** | **19.0** | **52.3** | **38.8** |

*Table 3.* Performance Comparison of GeoPixel in Referring Expression Segmentation on RRSIS-D dataset. The segmentation accuracy based on referring expressions is expressed through the Precision at IoU threshold of 0.5 (P@0.5), Overall Intersection-over-Union (oIoU) and Mean Intersection-over-Union (mIoU).

| METHOD | VALIDATION SET | | | TEST SET | | |
|---|---|---|---|---|---|---|
| | P@0.5 | oIoU | mIoU | P@0.5 | oIoU | mIoU |
| RRN (LI ET AL., 2018) | 51.09 | 66.53 | 46.06 | 51.07 | 66.43 | 45.64 |
| CSMA (YE ET AL., 2019) | 55.68 | 69.68 | 48.85 | 55.32 | 69.39 | 48.54 |
| LSCM (HUI ET AL., 2020) | 57.12 | 69.28 | 50.36 | 56.02 | 69.05 | 49.92 |
| CMPC (HUANG ET AL., 2020) | 57.93 | 70.15 | 50.41 | 55.83 | 69.22 | 49.24 |
| BRINet (HU ET AL., 2020) | 58.79 | 70.73 | 51.14 | 56.90 | 69.88 | 49.65 |
| CMPC+ (LIU ET AL., 2022) | 59.19 | 70.14 | 51.41 | 57.65 | 68.64 | 50.24 |
| LGCE (YUAN ET AL., 2024) | 68.10 | 76.68 | 60.16 | 67.65 | 76.34 | 59.37 |
| LAVT (YANG ET AL., 2024) | 69.54 | 77.59 | 61.46 | 69.52 | 77.19 | 61.04 |
| RMSIN (LIU ET AL., 2024C) | 74.66 | 78.27 | 65.10 | 74.26 | 77.79 | 64.20 |
| GEOPIXEL-FT | **80.00** | **81.77** | **67.99** | **83.33** | **84.90** | **67.30** |

### 5.3. Results

**Remote Sensing Grounded Conversation Generation:** A comparative analysis of the performance of various models on the RS-GCG task is evaluated across different metrics, including CIDEr, METEOR, CLAIR (Chan et al., 2023), AP50, mIoU, and recall, segmented into Uni-Target, Multi-Target, and Overall categories.CLAIR is an LLM-based similarity metric (GPT-4o in our case) that better aligns with human judgments. GeoPixel demonstrates superior performance in all metrics compared to the baselines, showing better fluency and text relevance in textual outputs while maintaining strong performance in more complex multi-target scenarios. In contrast, LISA† struggles with segmentation-based tasks, as evidenced by its low AP50 scores in all categories. PixelLM† shows a moderate improvement over LISA†, benefiting from better image feature scaling and segmentation token adjustments. GLaMM-ft exhibits improved outcomes due to a dedicated grounding encoder and GCG pre-training, however, its performance remains inferior to that of GeoPixel (as detailed in Table 2). Qualitative results are presented in Figure 4.

**Referring Remote Sensing Image Segmentation:** This task focuses on segmenting specific regions in aerial

*Table 4.* Performance Comparison on Referring Expression Detection Task, reporting Acc@0.5 / Acc@0.7 across 3 categories: unique, non-unique and overall. GeoPixel demonstrated substantial gains over existing methods in referring expression detection/localization.

| MODEL | UNIQUE | | NON-UNIQUE | | OVERALL | |
|---|---|---|---|---|---|---|
| | ACC@0.5 | ACC@0.7 | ACC@0.5 | ACC@0.7 | ACC@0.5 | ACC@0.7 |
| MINIGPT-v2 | 40.7 | 18.9 | 32.4 | 15.2 | 35.8 | 16.8 |
| LLAVA-1.5 | 51.1 | 16.4 | 34.8 | 11.5 | 41.6 | 13.6 |
| MINI-GEMINI | 41.1 | 9.6 | 22.3 | 4.9 | 30.1 | 6.8 |
| GEOCHAT | 57.4 | 22.6 | 44.5 | 18.0 | 49.8 | 19.9 |
| GEOPIX | 57.0 | 22.7 | 44.8 | 18.2 | 49.8 | 20.0 |
| GEOPIXEL-FT | **70.37** | **41.54** | **65.80** | **40.32** | **67.70** | **40.83** |

imagery guided by textual descriptions. The input prompt used is: `"Could you provide a segmentation mask for {referring_expression} in this image?"` The model generates the response, `"Sure, it is <SEG>."` where the corresponding embeddings of `<SEG>` token is subsequently decoded to produce the segmentation mask. To address this task, we fine-tune the GeoPixel model on the RRSIS-D (Liu et al., 2024c) dataset. The resulting GeoPixel-ft model demonstrates superior performance compared to recent approaches, as shown by results on the RRSIS-D test and validation sets in Table 3. The qualitative results are provided in Figure 5.

**Referring Remote Sensing Image Detection:** We formulate referring expression detection (RED) as a post-processing task on GeoPixel's predictions, deriving horizontal (HBBs) and oriented bounding boxes (OBBs) from output masks, for consistent evaluation. Table 4 compares performance on VRSBench (Li et al., 2024a), a dataset based on DOTAv2 and DIOR. To prevent data leakage, GeoPixel (trained on GeoPixelD) is finetuned on VRSBench without using any data from RRSIS-D (DIOR-based dataset). Moreover, GeoPixelD and VRSBench use DOTA's training set for training and validation set for testing. We further compare GeoChat and GeoPixel using OBBs (Table 5) with both models finetuned on VRSBench, highlighting the importance of pixel alignment in LMMs. Figure 10 illustrates the qualitative results.

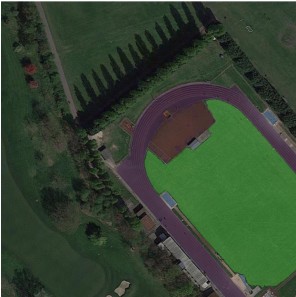 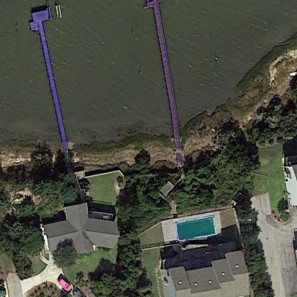 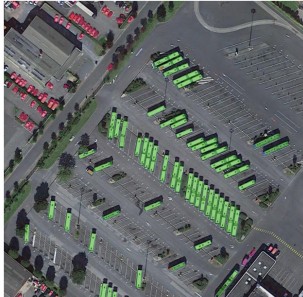 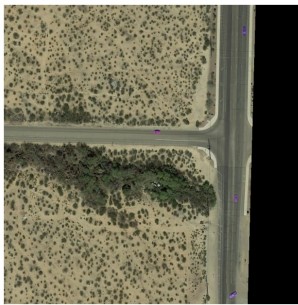

The image is a view of a sports complex, featuring a running track with a red running surface and green surrounding areas that may be grassy fields or additional sports facilities. The ground track field at the center is a well defined athletic track with a curved shape, surrounded by a grassy area with trees, and the soccer field at the bottom right is a well maintained grassy area with visible markings for gameplay. The structures, which could be seating areas or other facilities, are visible on one side of the track. The presence of trees and open spaces suggests that the complex is designed for outdoor activities and possibly community events, and the serene atmosphere is due to the absence of people in the scene.

The image is an aerial photograph of a residential area with several houses surrounded by trees. It features two prominent docks extending into a body of water, suggesting proximity to a lake or river. The layout of the roads and the positioning of the houses indicate a suburban setting, with a pier at the top being elongated, straight, and extending into the water with a perpendicular docking area at its end. A pier at the top is elongated, straight, and extends from the land into the water, with no visible structures or objects on it. A swimming pool at the bottom is rectangular, filled with blue water, and surrounded by a dark colored deck. A solitary small vehicle is parked on a driveway at the bottom left of the image, adjacent to a house with a dark roof. The presence of greenery and the absence of commercial buildings or high density housing structures suggest that this is a quiet, possibly affluent neighborhood.

The image is an aerial view of a parking lot with numerous cars parked in designated spaces, arranged in orderly rows, indicating a well organized parking system. The parking lot appears to be part of a larger facility, possibly a commercial or industrial complex, as suggested by the presence of trees and other structures. The image depicts a large parking area with multiple large vehicles, including buses and possibly coaches, parked in an organized manner. There are multiple small vehicles scattered across various regions. The absence of people in the image could imply that the photo was taken during a time of low activity or from a high vantage point where individuals are not easily discernible.

The image is an aerial photograph of a rural area with a road cutting through it, appearing to be a two lane highway with vehicles traveling on it. The surrounding landscape is predominantly dry and sparsely vegetated, indicative of a desert or arid environment. On the road, there are four small vehicles. The scene has a natural and undeveloped appearance, with no visible buildings or infrastructure other than the road itself.

*Figure 4.* Qualitative results of GeoPixel on RS-GCG. Contextually rich descriptions of RS imagery with grounded object annotations. Depending on object scale and density, it employs instance masks for precise delineation of individual objects (right and middle-right images) while semantic masks capture broader categories, such as large clusters of vehicles or small objects (middle-left and left images).

*Table 5.* Visual grounding performance on VRSBench dataset using oriented bounding boxes for referring object localization. The observed performance improvement underscores the critical role of precise pixel-level alignment.

| MODEL | UNIQUE | | NON-UNIQUE | | OVERALL | |
|---|---|---|---|---|---|---|
| | Acc@0.5 | Acc@0.7 | Acc@0.5 | Acc@0.7 | Acc@0.5 | Acc@0.7 |
| GEOCHAT | 32.3 | 12.6 | 18.5 | 5.7 | 24.3 | 8.6 |
| GEOPIXEL-FT | **54.48** | **24.87** | **60.51** | **30.97** | **58.00** | **28.42** |

### 5.4. Ablation Study

**Inference Resolution Effect:** Increasing inference patches improves all evaluation metrics, reflecting better comprehension of visual content (Table 6). At $\mathcal{P} = 9$, CIDEr increases from 14.6 to 20.5, and METEOR improves from 23.1 to 24.3, indicating enhanced semantic understanding as image tokens scale up. The moderate mAP and mIoU gains suggest that while HR inference contributes to superior localization accuracy, competitive performance can still be maintained at lower resolutions when the model is pretrained at HR. The superior results at ($\mathcal{P} = 9$) underscore the critical role of incorporating fine-grained spatial details during the training phase for generalized feature learning.

**Annotation Complexity Effect:** GeoPixel adjusts its masking based on object size and distribution (Figure 4), using instance masks for precise identification, while semantic masks for broader categories (clusters/small objects). In scenarios requiring both granularity and generalization, the model employs hybrid annotations, blending instance-level

*Table 6.* Effect of Inference Resolution. Reported metrics show the relationship between resolution and overall performance.

| TRAINING PATCHES | INFERENCE PATCHES | CIDER | METEOR | AP50 | mIoU | RECALL |
|---|---|---|---|---|---|---|
| $\mathcal{P} = 9$ | $\mathcal{P} = 1$ | 14.6 | 23.1 | 12.9 | 47.8 | 32.2 |
| | $\mathcal{P} = 4$ | 17.7 | 23.9 | 16.6 | 51.8 | 37.1 |
| | $\mathcal{P} = 9$ | **20.5** | **24.3** | **17.6** | **52.1** | **37.4** |

*Table 7.* Effect of Annotation Complexity. Avg. Len is the average character length of captions.

| DATA | OBJECTS | PHRASES | AVG. LEN | mIoU | RECALL |
|---|---|---|---|---|---|
| INSTANCES ONLY | 1,740 | 1,740 | 634 | **58.4** | **48.8** |
| SEMANTIC ONLY | 21,483 | 698 | 518 | 44.1 | 37.7 |
| MIX DATA | 38,161 | 2,989 | 737 | 50.9 | 33.3 |

and semantic mask representations(as seen in Figure 1). Table 7 shows that this annotation complexity impacts performance, with mixed annotations yielding the lowest recall.

Remote sensing images often contain visually similar objects with subtle variations in appearance, spatial layout, and proximity, yet exhibit significant scale variations across images. This complexity challenges the model's ability to distinguish between object presence, quantity, and the appropriate annotation type required (e.g., instance or semantic). The challenge is pronounced in the semantic-only category, where the model exhibits the lowest mIoU scores, indicating two key challenges: the model's ability to ensure full coverage of all instances within a category and grouping

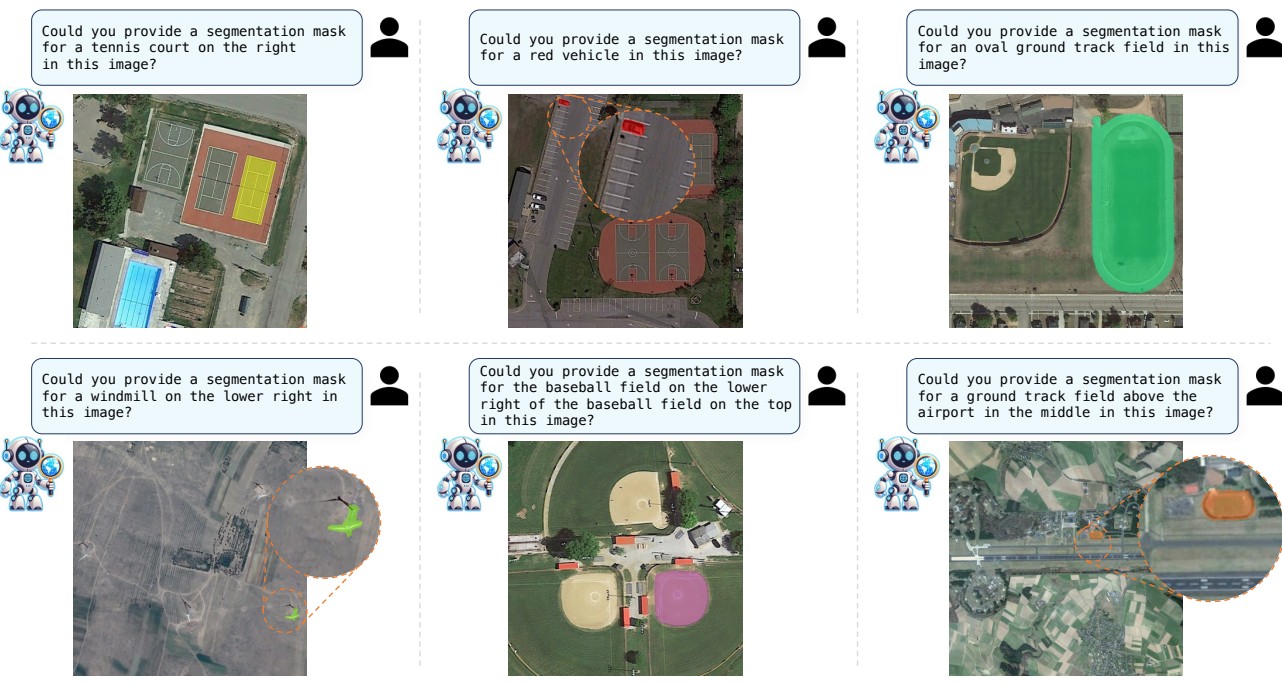

*Figure 5.* Qualitative results of GeoPixel's capability in referring remote sensing expression segmentation. The figure highlights Geopixel's ability to interpret referring expressions of varying lengths and generate precise segmentation masks, adapting to scale variations, as shown in the ground track fields. Spatial descriptors (e.g "right", "lower right"), and object characteristics (e.g "red") are interpreted with precision to achieve accurate segmentation.

*Table 8.* Effect of Data Complexity and Training Vision Projection (VP) Layer. T stands for Trainable and F for Frozen.

| TRAINING DATA | | VP | CIDER | METEOR | AP50 | mIoU | RECALL |
|---|---|---|---|---|---|---|---|
| SET-1A | SET-1B | | | | | | |
| ✓ | | T | 19.3 | 23.6 | **18.2** | 48.0 | 33.6 |
| ✓ | ✓ | T | **20.5** | 24.0 | 17.8 | **51.7** | **36.7** |
| ✓ | ✓ | F | 18.7 | **24.4** | 15.3 | 51.6 | 35.1 |

objects under a unified mask instead of individual instances. The low mask recall in mixed data further suggests that generalizing masking decisions in dense scenes is difficult due to scale and distributional variability of geospatial entities.

**Role of Data Complexity:** Table 8 compares GeoPixel's performance across data partitions with varying masking complexity. Set-1A is less complex, with no intraclass segmentation differences. Each instance of a single class is individually masked or represented using a semantic mask. Set-1B introduces a higher complexity by assigning instance masks to larger objects while grouping smaller ones under a semantic mask within the same class. For example, larger boats may be individually described, while smaller boats in the image could be grouped under a single semantic description. This ablation evaluates the model's ability to handle varying levels of annotation granularity, providing

insights into its ability to generalize across different scales and segmentation strategies. The results indicate that the inclusion of more complex annotation (Set-1B) enhances segmentation accuracy and descriptive detail, as the model is trained with more diverse mask configurations.

**Vision Projection:** Next we study the effect of training the vision projection layer by comparing the performance when the vision projection layer is fixed or trainable during the fine-tuning stage. Table 8 summarizes the results. Training the vision projection layer results in an improvement in some metrics, highlighting the role of feature alignment.

## 6. Conclusion

We present GeoPixel, a large multimodal model (LMM) designed specifically for the unique challenges of high-resolution remote sensing (RS) image analysis. GeoPixel introduces a robust end-to-end architecture capable of adaptive image partitioning and pixel-level grounding, enabling the precise interpretation and generation of geospatially aware descriptions in RS imagery. By addressing key limitations of current LMMs, such as low-resolution constraints and coarse object-grounding, GeoPixel provides a fine-grained visual understanding that bridges the gap between language and high-resolution RS data.

## Impact Statement

GeoPixel, a large multimodal model, is designed to enhance the fine-grained spatial understanding of high-resolution remote sensing (RS) imagery. Our work contributes to the advancement of machine learning by improving pixel-level object grounding, a critical capability for geospatial applications such as urban planning, environmental monitoring, disaster response, and infrastructure assessment.

From an ethical perspective, GeoPixel operates on publicly available remote sensing datasets, ensuring transparency and reproducibility in research. However, the use of high-resolution imagery raises privacy considerations, particularly in applications that involve urban environments. Although our model does not process personally identifiable information (PII) or real-time surveillance data, we acknowledge that future extensions incorporating private datasets should adhere to strict data governance and privacy-preserving policies.

Regarding social impact, the ability of GeoPixel to generate detailed geospatially grounded descriptions can help in disaster relief efforts by improving situational awareness from satellite imagery. It can also improve climate change monitoring by facilitating precise land use classification and deforestation tracking. We emphasize that GeoPixel is developed for scientific and humanitarian purposes and encourage ethical deployment in accordance with open research principles. Overall, this work aligns with the broader goal of advancing machine learning in remote sensing, promoting open and responsible AI applications, and fostering positive societal impact while remaining mindful of potential ethical considerations.

## Acknowledgement

The computations were enabled by resources provided by NAISS at Alvis, partially funded by the Swedish Research Council through grant agreement no. 2022-06725, LUMI hosted by CSC (Finland) and LUMI consortium, and by Berzelius resource provided by the Knut and Alice Wallenberg Foundation at the NSC

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

## A. Limitations and Challenges

While GeoPixel has demonstrated significant advances in pixel-level grounding for high-resolution RS images, several challenges remain. These challenges are particularly evident in the following failure cases (illustrated in Figure 6). The model occasionally produces erroneous masks due to ambiguities in the masking strategy, particularly in determining object presence and quantity, as well as deciding whether semantic segmentation or instance-level annotation is appropriate. An incorrect decision in this regard can result in repetitive descriptions of visually similar objects, leading to inconsistencies in the generated output. Furthermore, such errors may manifest as fragmented or overlapping masks, introducing confusion in object delineation and undermining the overall segmentation quality. Moreover, the model often confuses instance masks within the same spatial location, particularly in densely populated or crowded images.

Future work may focus on addressing these challenges by incorporating more robust masking strategies and dynamic resolution adjustment techniques to improve segmentation accuracy in complex scenes. Additionally, extending GeoPixel's capabilities to integrate multimodal data, such as Synthetic Aperture Radar (SAR) or infrared imagery, could significantly enhance its ability to analyze diverse remote sensing datasets. GeoPixel is a significant step forward in leveraging the potential of LMMs for remote sensing, opening new avenues for research in this critical domain.

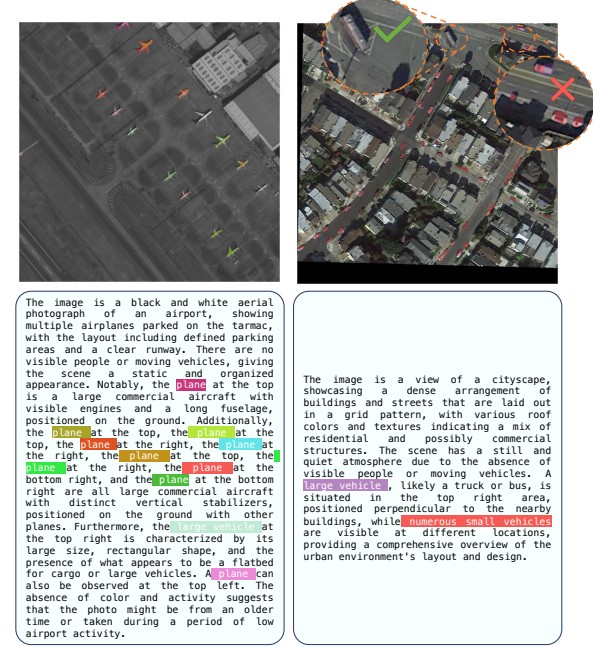

*Figure 6.* Failure case due to incorrect mask association (left) and wrong instance segmentation in the same spatial region (right).

## B. GeoPixelD dataset

**Preprocessing and Marking**: We utilize the instance-level annotated dataset, iSAID (Waqas Zamir et al., 2019), to generate grounded conversations through our annotation pipelines. The images undergo a preprocessing step in which they are cropped into 800 x 800 pixel patches. Objects for instance annotations are selected based on an area threshold to ensure their reasonable size, therefore preventing the marker from obscuring a significant portion of the object and maintaining its distinguishability. A 14 x 14 pixels fixed size marker is used, regardless of the actual dimensions of the object. However, the marker's placement is determined based on the segmentation mask's area and shape. For large objects, the marker is positioned at the center of the mask if the calculated center falls within the mask boundaries; otherwise, it is adjusted to the nearest point on the object's border. For small objects, the center of the bounding box is aligned with a point on the polygon mask boundary, which typically results in an average marker overlap of 50% with the object.

In addition, multiple marking techniques were explored, including bounding boxes, masks, contours, and numerical markers, to determine their impact on accuracy and object fidelity. Our findings reveal that bounding boxes and contours add superfluous visual information that can obscure the fine details of the object. In contrast, simple numerical markers placed

directly on the object effectively signal its presence without compromising visual clarity or introducing noise, thereby preserving the integrity of object details for enhanced model performance.

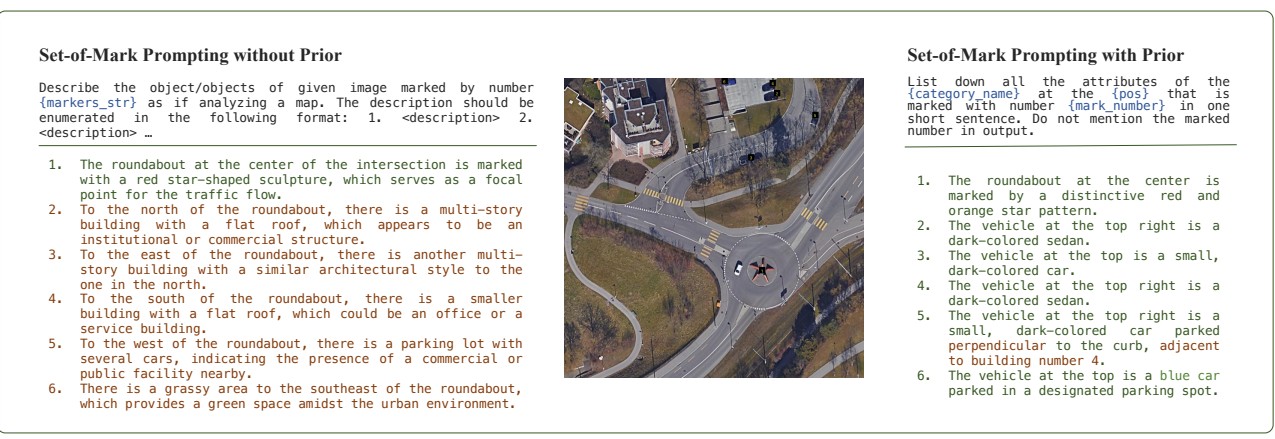

*Figure 7.* Comparative effectiveness of SOM prompting methods, highlighting the critical role of priors. Without priors, SOM relies solely on the VLM to detect and describe marked objects independently, resulting in inaccurate descriptions and hallucinated markers in complex remote sensing scenes. In contrast, SOM with priors utilizes explicit marker positions ({pos}) and predefined object categories ({category_name}) as priors, providing structured prompts that reduce ambiguity and guide the VLM to produce precise and reliable descriptions. Incorrect parts are noted in red whereas correct parts are noted in green.

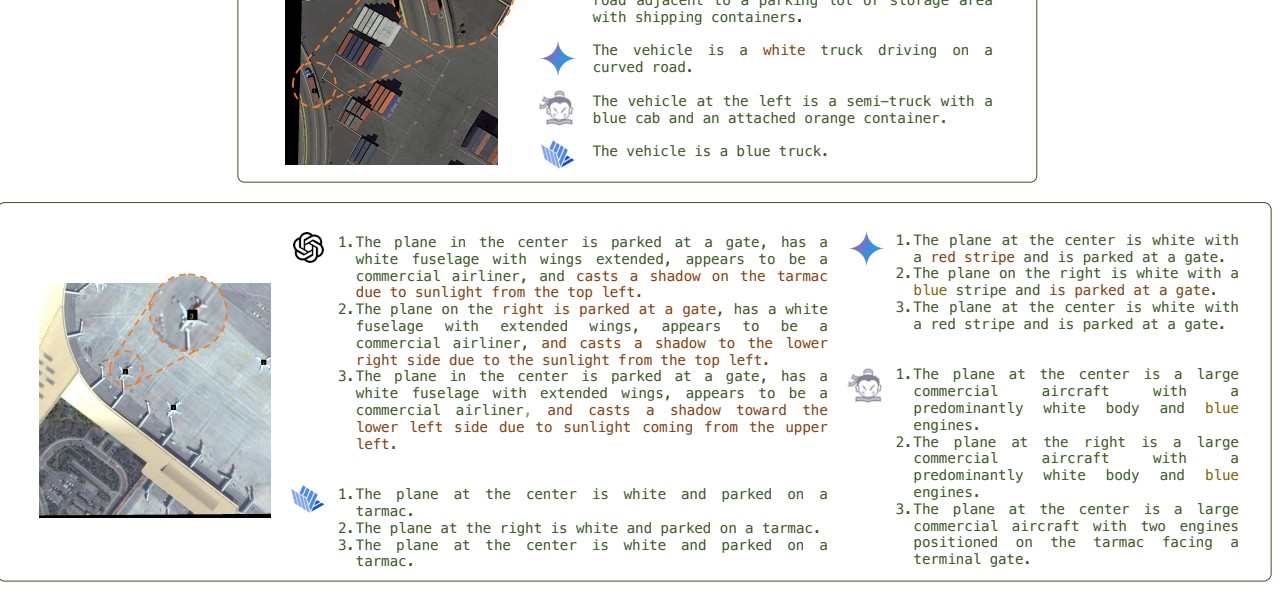

*Figure 8.* Comparison of open-source and proprietary models for prior-informed set of marks (SOM) prompting for RS imagery. Incorrect parts are noted in red whereas correct parts are noted in green.

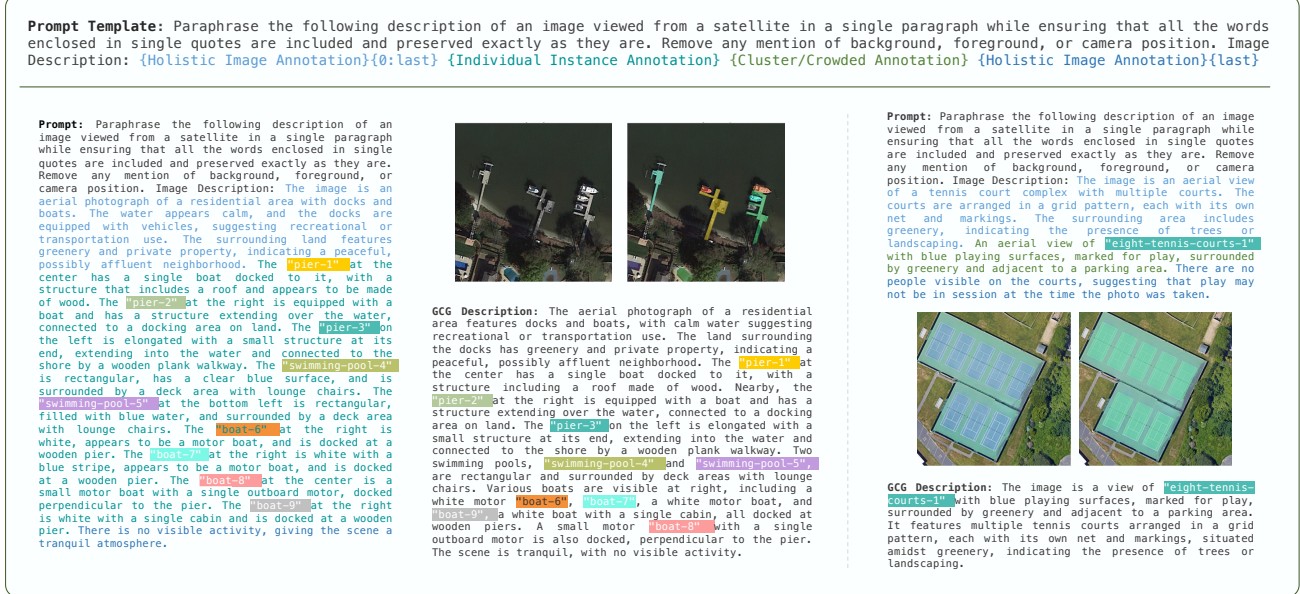

*Figure 9.* Unifying Annotations through LLM Paraphrasing and Text Marking to track associated masks. Objects are indexed numerically (e.g., "object-N"), and holistic (blue), individual (teal), and cluster (green) annotations are concatenated into a single image description. Paraphrasing instructions with combined description produce a concise, consistent GCG description that eliminates redundancy while preserving object-mask associations, even with reordering.

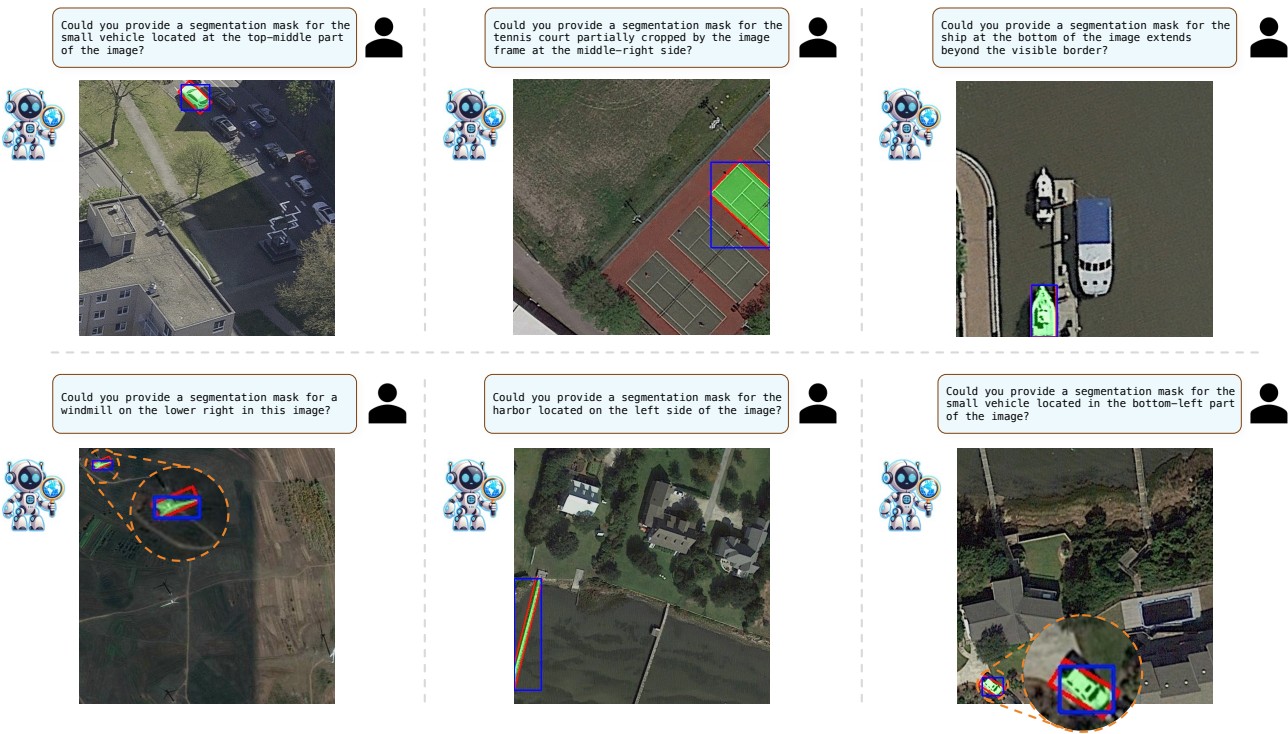

*Figure 10.* Qualitative results of GeoPixel's capability in Referring Expression Detection on VRSBench. Segmentation masks generated by GeoPixel are used to derive horizontal bounding boxes (shown in blue) and oriented bounding boxes (shown in red).

