# OpenReview forum: "GeoPixel: Pixel Grounding Large Multimodal Model in Remote Sensing"
_ICML.cc/2025/Conference — ICML 2025 poster_

### Official Review · Reviewer_fxwn · 2025-03-08

**Overall Recommendation:** 4

**Summary:**

The paper introduces GeoPixel, a remote sensing multimodal LLM for pixel level understanding and reasoning in high resolution aerial images. The authors present GeoPixelD, a new dataset with detailed, spatially-aware annotations for grounded conversations in remote sensing. The authors develop an adaptive image divider module for high resolution image understanding within the MLLM. Experimental results demonstrate GeoPixel's superior performance in generating grounded conversations and segmenting referred objects in remote sensing data, highlighting its advancements in understanding and interpreting visual information.

**Claims And Evidence:**

1. GeoPixel is designed to handle and reason about high resolution aerial images which is confirmed by performance comparison with models like PixelLM and RSMIN.
2. GeoPixel is designed to provide pixel grounded outputs and its capability is demonstrated on the RS-GCG dataset.
3. Although the impact statement mentions that GeoPixel can be used for urban planning and disaster response, the paper does not showcase any such applications. Instead, it focuses solely on demonstrating the performance of GeoPixel on benchmark datasets.

**Essential References Not Discussed:**

TeoChat: Irvin, J. A., Liu, E. R., Chen, J. C., Dormoy, I., Kim, J., Khanna, S., ... & Ermon, S. (2024). Teochat: A large vision-language assistant for temporal earth observation data. ICLR.

**Experimental Designs Or Analyses:**

The experiments shown in the paper are adequate to evaluate the pixel-level understanding and reasoning capability of GeoPixel.

**Methods And Evaluation Criteria:**

1. There is no comparison of GeoPixel shown against other remote sensing based MLLM such as GeoChat, SkyEyeGPT or TeoChat in tasks like remote sensing image captioning or object detection. Although they are mentioned in the related works and Table-1, it might be interesting to see a comparison with these models.

**Other Comments Or Suggestions:**

NA

**Other Strengths And Weaknesses:**

1. The paper introduces GeoPixelID, a dataset for grounded conversation in remote sensing. The paper also introduces a benchmark dataset to evaluate RS-LLM models on the task of referring segmentation.

**Questions For Authors:**

NA

**Relation To Broader Scientific Literature:**

GeoPixel is one of the early works in remote sensing that developed a MLLM for grounded conversations and pixel-level reasoning. While such models exist for other domains, GeoPixel outperforms these generalist models, particularly for high-resolution aerial images. As mentioned in the impact statement, GeoPixel can be utilized in remote sensing applications like urban planning and disaster response (although not shown in the paper).

**Theoretical Claims:**

NA

---

> ### Author Rebuttal · Authors · 2025-04-01
>
> Dear Reviewer fxwn,
>
> Thank you for your comprehensive review of our submission. We appreciate your insights and the opportunity to clarify and expand upon aspects of our work.
>
> **Performance Benchmarking:** We appreciate your acknowledgment of GeoPixel's superior performance in handling high-resolution aerial images compared to models like PixelLM and RSMIN. Regarding your suggestion to compare GeoPixel with other multimodal models in remote sensing (e.g., GeoChat, SkyEyeGPT, TeoChat), we would like to clarify that GeoPixel is specifically designed as a segmentation-centric model. In contrast, current RS multimodal language models do not support segmentation outputs .
>
> To enable a fair comparison, we formulated referring expression detection as a post-processing task over GeoPixel's segmentation predictions. Specifically, the segmentation masks generated by GeoPixel are used to derive horizontal bounding boxes (HBBs) and oriented bounding boxes (OBBs), providing a consistent evaluation framework across models. This approach was applied on VRSBench, allowing us to benchmark GeoPixel against a range of existing models under comparable detection metrics.
>
> GeoPixel demonstrated substantial gains over existing methods in referring expression detection/localization, measured by Accuracy\@0.5 and Accuracy\@0.7 across unique, non-unique, and overall categories:
>
> **MiniGPT-v2** scored 40.7/18.9 (unique), 32.4/15.2 (non-unique), and 35.8/16.8 overall.
>
> **LLaVA-1.5** achieved 51.1/16.4 (unique), 34.8/11.5 (non-unique), and 41.6/13.6 overall.
>
> **Mini-Gemini** showed lower performance with 41.1/9.6 (unique), 22.3/4.9 (non-unique), and 30.1/6.8 (overall).
>
> **GeoChat** reached 57.4/22.6 (unique), 44.5/18.0 (non-unique), and 49.8/19.9 (overall).
>
> **GeoPix** performed similarly with 57.0/22.7 (unique), 44.8/18.2 (non-unique), and 49.8/20.0 (overall).
>
> **GeoPixel** achieved the best results: **70.37/41.54** (unique), **65.80/40.32** (non-unique), and **67.70/40.83** (overall).
>
> We further compared **GeoChat** and **GeoPixel** on VRSBench using orientated bounding boxes for referring expression detection.
>
> **GeoChat** scored 32.3/12.6 (unique), 18.5/5.7 (non-unique), and 24.3/8.6 (overall).
>
> **GeoPixel** significantly outperformed, achieving **54.48/24.87** (unique), **60.51/30.97** (non-unique), and **58.00/28.42** (overall).
>
>
> **Relation to Broader Scientific Literature:** Thank you for acknowledging GeoPixel's contribution as an early work in remote sensing MLLMs. It sets a foundation for subsequent research in this rapidly evolving domain. Also, we appreciate your reference to the concurrent work TeoChat. We will discuss it in the revised manuscript to contextualize GeoPixel's contributions within the broader scientific dialogue.
>
>
> **Further Contributions:** Your recognition of the GeoPixelD dataset as a valuable resource for the community is encouraging. We believe that both GeoPixelD will help propel forward the capabilities of RS-MLLM models.
>
> We are preparing a revision that incorporates these insights and will provide additional data and qualitative results to address the gaps you've highlighted. Thank you once again for your thorough evaluation and constructive criticism.

---

> > ### Comment · Reviewer_fxwn · 2025-04-07
> >
> > I thank the authors for the rebuttal. I have gone through the concerns raised by other reviewers. I believe the paper fits well into the application driven ML track and hence I would like to maintain my original rating.

---

### Official Review · Reviewer_Vj3H · 2025-03-09

**Overall Recommendation:** 3

**Summary:**

This work introduces a multi-modal Grounded Conversation Generation (GCG) dataset that includes grounded descriptions for high-resolution remote sensing images, along with a benchmark featuring human-verified annotations. By leveraging recent advances in large vision-language models, the authors propose a model aimed at achieving fine-grained visual understanding in remote sensing imagery.

**Claims And Evidence:**

Method novelty. The method framework has no significant difference with existing MLLMs, such as LISA and VisionLLM-v2. The idea of dividing the input images into local and global regions is widely explored in large vision-language models, such as Llava-Next.

**Essential References Not Discussed:**

Some Remote Sensing (RS) LMMs are missing in the related work section, such as, SkyEyeGPT, VRSBench, Popeye, MMM-RS, and RS-MoE.

**Ethical Review Concerns:**

Annotation Accuracy: The reliance on automatically generated annotations by vision-language models may lead to inaccuracies, even when using state-of-the-art proprietary models.

**Ethical Review Flag:**

Flag this paper for an ethics review.

**Ethics Expertise Needed:**

["Responsible Research Practice (e.g., IRB, documentation, research ethics, participant consent)"]

**Experimental Designs Or Analyses:**

The experimental design is mostly good, except for minor unfairness in Table 3.

**Methods And Evaluation Criteria:**

The proposed method and evaluation make sense for the target problem.

**Other Comments Or Suggestions:**

No further suggestions.

**Other Strengths And Weaknesses:**

1. Method Novelty
The framework does not significantly differ from existing MLLMs, such as LISA and VisionLLM-v2. The strategy of partitioning input images into local and global regions is a well-explored concept in large vision-language models (e.g., Llava-Next).

2. Dataset Collection
Grid-Based Annotation: In the individual pipeline, images are divided into 3×3 grids, with annotations generated for each grid. This approach raises concerns when objects are positioned at grid boundaries or span across multiple grids, potentially leading to incomplete or inaccurate annotations.
Annotation Accuracy: The reliance on automatically generated annotations by vision-language models may lead to inaccuracies, even when using state-of-the-art proprietary models.

**Questions For Authors:**

Pretraining Discrepancy: In Table 3, the proposed GeoPixel model is pretrained on the iSAID-based GeoPixelD dataset, whereas the comparison methods are trained solely on the target dataset, which creates an imbalance in the experimental setup.
Evaluation Metric: For the detailed captioning task, the use of CIDER as the evaluation metric may not be entirely appropriate. This limitation should be discussed in the manuscript.

**Relation To Broader Scientific Literature:**

This work lies in the scope of MLLMs in remote sensing. There are already many attempts in this direction, with only a few focusing on pixel-level tasks. This work mainly differs in the newly collected grounded detailed dataset.

**Theoretical Claims:**

The work does not include any theoretical proofs.

---

> ### Author Rebuttal · Authors · 2025-04-01
>
> Dear Reviewer Vj3H,
>
> Thank you for your thorough review and insightful comments regarding our submission. We appreciate the opportunity to address the issues raised and clarify aspects of our research methodology and dataset.
>
> **Method Novelty and Framework Comparison:** While our method framework resembles existing MLLMs like LISA and VisionLLM-v2, our approach is specifically tailored to the unique challenges of remote sensing (RS) imagery. The novelty lies in integrating high-resolution image comprehension with pixel-level grounding capabilities. This cohesion is critical for RS, as it enhances the model's capacity to decipher complex visual information at a granular level, thereby providing a more refined understanding and interaction with diverse geospatial objects. Moreover, our data annotation pipeline is uniquely designed for RS imagery, leveraging spatial priors and region-specific markers to extract and represent regional information effectively.
>
> **Dataset Collection and Annotation Methodology:** Thank you for raising concerns about annotation accuracy with 3×3 grids. To address this, we would like to clarify the robustness of our dataset annotation methodology: In instance annotation, precise object assignment is ensured through a pixel overlap ratio, rather than just bounding box or object centers, to ensure accurate placement of objects. Furthermore, in scenarios where ambiguity arises, each individual annotation is already supplemented with a distinct set of markers. These markers aid in distinguishing between instances, particularly when locational ambiguity exists. Further reinforced through the integration of category priors, this proves to be a highly effective strategy for instance-specific annotations. In comparison, situations involving cluster annotations where markers are absent, we have adopted an enhanced localization mechanism with a multi-grid hierarchical system. This adjustment significantly mitigates ambiguities by refining the spatial grid at which annotations are applied, ensuring comprehensive and precise coverage across intersecting grids.
>
> **Pretraining Discrepancy and Evaluation Metrics:** Regarding the pretraining discrepancy mentioned, I would like to clarify that all the models in Table 3 are trained on GeoPixelD data under the same training conditions, parameters, and epochs. Evaluation Metric: We acknowledge that CIDEr may not fully capture the nuances of detailed image captioning. To address this, we conducted additional evaluations using the CLAIR score [1], an LLM-based metric (GPT-4o in our case) that better aligns with human judgments. The results are GLaMM: 43.11, LISA+: 68.96, PixelLM+: 73.93, GLaMM-ft: 71.74, and GeoPixel **77.50**. Notably, GeoPixel outperforms all other models in this evaluation and across ROUGE-1, ROUGE-2, and ROUGE-L scores.  [1] Chan, D., Petryk, S., Gonzalez, J. E., Darrell, T., & Canny, J. (2023). CLAIR: Evaluating image captions with large language models. arXiv preprint arXiv:2310.12971.
>
> **Example of CLAIR score and reasoning:** "score": 0.25, "explanation": "The candidate and reference sets both describe an aerial view involving docks and a body of water with a dark appearance. However, there are notable discrepancies in the details: the candidate mentions four piers with specific structures on them, while the reference describes only two piers. The description of the boats also differs, with the candidate mentioning a solitary boat moored at a dock, while the reference describes two boats in different positions."
>
> **Ethical Review and Annotation Accuracy:** In addressing ethical review concerns regarding the accuracy and use of automatically generated annotations, it is important to emphasize the rigorous validation process implemented in this research. We employed a robust validation protocol where each gcg description in the test set was meticulously verified by expert annotators. These specialists, trained in recognizing and correcting discrepancies in automated text, manually corrected any data discrepancies found. Moreover, training set annotations were also rigorously filtered to eliminate aerial perspective inconsistencies, artifacts such as marker identifiers, fore/background references, depth cues, and inconsistent descriptors. This validation process underscores our commitment to ethical research practices.
>
> **Related Work:** We thank you for pointing out several relevant works. We will update our manuscript to include discussions on RS-specific LMMs like SkyEyeGPT, VRSBench, Popeye, MMM-RS, and RS-MoE, providing a more comprehensive overview of the landscape and situating our contributions within it.
>
> We hope that these clarifications and planned improvements address your concerns and demonstrate our commitment to advancing the field of RS through responsible and innovative research practices.  Thank you for your constructive feedback, invaluable in refining our work.

---

> > ### Comment · Reviewer_Vj3H · 2025-04-07
> >
> > The author's response has addressed part of my concerns. First, regarding data quality, this statement should be added to the data collection part: "We employed a robust validation protocol where each gcg description in the test set was meticulously verified by expert annotators", along with details of the validation protocol. Second, even though the designed method is technically sound, it is mainly built from existing techniques, and the novelty is still questionable. Overall, I'll improve my score to weak accept, given the dataset contribution.

---

### Official Review · Reviewer_zikq · 2025-03-12

**Overall Recommendation:** 2

**Summary:**

This paper introduces GeoPixel and GeoPixelD.

GeoPixel is a combination of models that receives high-res RGB satellite imagery and outputs text (e.g., a description in natural language) and a dense segmentation map. The combination consists of a frozen vision encoder that "tokenizes" the image and feeds these tokens to a pretrained LLM. The LLM outputs text, which is fed to a pixel decoder along with visual tokens. And the pixel decoder outputs the segmentation map. GeoPixel performs well on both image captioning / describing (in natural language) and image segmentation.

GeoPixelD is a dataset that consists of matching high-res RGB satellite imagery and natural language descriptions, constructed via a semi-automated pipeline.

**Claims And Evidence:**

Yes. The paper claims to outperform the SOTA on text-guided image segmentation and image description tasks. I believe the results on page 7 support this.

**Essential References Not Discussed:**

This paper does not discuss RS-specific foundation models, such as SatMAE, etc.

**Experimental Designs Or Analyses:**

Overall, the experiments look sound. According to the end of section 5.2, all LMM models are finetuned on GeoPixelD. However, in table 2, GLaMM has two entires: "GLaMM" and "GLaMM-FT". LISA and PixelLM do not have the "FT" suffix, were these _not_ finetuned on GeoPixelD?

**Methods And Evaluation Criteria:**

There are many RS benchmarks that this paper does not use to evaluate GeoPixel. I believe this is because GeoPixel is designed for two specific tasks (RS image descriptions in natural language and text-guided image segmentation) for which few datasets exist.

Can the authors please clarify why they did not evaluate GeoPixel on other RS image to text datasets, like those used in GeoChat, EarthGPT, etc.?

**Other Comments Or Suggestions:**

None.

**Other Strengths And Weaknesses:**

As discussed, GeoPixel is quite similar to GLaMM. I see the differences between GeoPixel and GLaMM as:

1. GeoPixel does not use an explicit region encoder (see top left of Figure 2 in GLaMM). Instead, GeoPixel tiles each high-res image and encodes them independently and globally (via downsampling).
2. GeoPixel uses PLoRA, which applies LoRAs to the vision tokens inside of the LLM.

To me, #1 makes sense in RS since we may not have a strong region encoder to extract salient regions. Tiling a high-res image seems like a sensible choice. #2 also seems reasonable to me, as the LLM likely needs to be adjusted a bit. However, I do not see ablations on these choices that empirically justify them.

Overall, I expect more experiments from an applied submission with limited technical novelty.

**Questions For Authors:**

Please see above.

**Relation To Broader Scientific Literature:**

GeoPixel demonstrates a method to leverage pretrained models as components in a system that can be finetuned successfully on another domain (remote sensing). This research question is relevant to many ML applications. That being said, the GeoPixel method itself is very similar to GLaMM (which was not applied to RS).

**Theoretical Claims:**

No theoretical claims.

---

> ### Author Rebuttal · Authors · 2025-04-01
>
> Dear Reviewer zikq,
>
> Thank you for your detailed review and insights on our submission, GeoPixel. We appreciate the opportunity to address your concerns and clarify aspects of our research.
>
> **Evaluation on Other RS Image-to-Text Datasets:** You raised an important question regarding why GeoPixel was not evaluated on additional RS image-to-text datasets, such as those used in GeoChat, EarthGPT, etc. The primary reason for this was the specific focus of GeoPixel on high-resolution satellite imagery coupled with text-guided image segmentation. While valuable, the datasets used in GeoChat EarthGPT often encompass varied tasks that do not align directly with the dense segmentation focus of GeoPixel. As the stated models do not support referring expression segmentation, we formulate referring expression detection as a post-processing task applied to the outputs of GeoPixel. Specifically, segmentation masks predicted by GeoPixel are utilized to derive horizontal bounding boxes (HBBs) and oriented bounding boxes (OBBs), enabling a consistent basis for comparative evaluation on VRSBench.
>
> GeoPixel outperforms GeoChat in referring expression detection in HBB and OBB, reporting Accuracy\@0.5 / Accuracy\@0.7 across unique, non-unique, and overall cases. The gain in performance is summarized as follows: HBB-based Detection Unique: **+12.97 / +18.94** Non-Unique: **+21.30 / +22.32** Overall: **+17.90 / +20.93** OBB-based Detection Unique: **+22.18 / +12.27**, Non-Unique: **+42.01 / +25.27**, Overall: **+33.70 / +19.82**.
> Both models were evaluated after finetuning on VRSBench; the gain in performance clearly shows the importance of pixel-level alignment in multimodal models.
>
> **Models Finetuning Clarification:** The models LISA+ and PixelLM+ were indeed modified and finetuned on the GeoPixelD dataset. The designation "FT" was specifically used for GLaMM to distinguish between its original and finetuned versions, as it was not modified. We will ensure that future iterations of the manuscript clearly articulate these distinctions to avoid confusion regarding the experimental setups and model adaptation status.
>
> **Methodological Choices and Justifications:** Typically employed for tasks like region-specific captioning, explicit encoders are not used in our approach. Instead, we utilize a Set of Marks (SOM) alongside spatial priors, as detailed in our data annotation pipeline, to target and delineate specific regions precisely. This method ensures accurate regional descriptions and functions independent of model frameworks and offers an effective alternative. Table 4 presents the ablation study on the impact of tiling high-resolution images, where P=1 denotes the use of a single image patch, effectively implying no tiling.
>
> **Relation to Broader Scientific Literature and Technical Novelty:** We appreciate the observation regarding the conceptual similarity between GeoPixel and models like GLaMM. However, remote sensing (RS) imagery demands high spatial resolution to capture fine structural details of ground objects. GLaMM's input resolution (336×336) is inadequate for representing RS data's rich spatial context and intricate features. GeoPixel addresses this by combining high-resolution visual understanding with pixel-level grounding, making it well-suited for the complexities of RS imagery. Additionally, our semi-automated data annotation pipeline, specifically tailored for RS imagery, extracts regional information through a set of marks and spatial priors, which also constitutes a key contribution.
>
> In the revised manuscript, we will discuss RS-specific foundation models (e.g., SatMAE) to contextualize our work further within the broader literature.
>
> We hope that these clarifications address your concerns adequately. Thank you for your constructive feedback, which is invaluable in improving our work.

---

> > ### Comment · Reviewer_zikq · 2025-04-07
> >
> > After reading the authors' rebuttal and other reviews, I keep my score a weak reject. There are two main reasons:
> >
> > 1. Limited technical novelty. There are two main innovations from GLaMM: Dividing images into tiles and using PLoRAs [1]. I'm not an expert in vision-language models (VLM), but I believe many of them can process multiple images / tiles in a single sample. Secondly, PLoRA has been used in (and was originally designed for) VLMs. Thus, the technical contributions are minor.
> >
> > 2. Limited experiments, specifically ablations. The submission has fewer experiments than I expect; however, as the authors address, there are very few public datasets for this specific task. Their method could be adapted to perform similar tasks, and the authors provided 1 such example in their rebuttal. However, the lack of public datasets does not explain the paper's lack of ablations. Ablations are crucial to help understand any proposed method. In response to my request for ablations, the authors pointed to Table 4 and did not address PLoRA. Table 4 varies the number of tiles at test time {1, 4, 9} but does not retrain the model under a different condition. For example, the model could be trained with 4 patches and also tested with {1, 4, 9} to test resolution extrapolation.
> >
> > Finally, since the two main technical contributions are not adequately ablated, it remains unclear how these two elements (and more!) contribute to GeoPixel's success. Thus, I cannot recommend this paper be accepted.
> >
> > [1] https://arxiv.org/abs/2401.16420

---

> > > ### Author Response · Authors · 2025-04-08
> > >
> > > We appreciate the reviewer’s comments and would like to clarify several key points regarding our work's novelty and experimental depth.
> > >
> > > **On Technical Novelty:**
> > >
> > > While some existing VLMs are capable of handling multiple tiles, GeoPixel introduces a tiling strategy in conjunction with pixel-level grounding, uniquely enhancing spatial understanding in high-resolution remote sensing (RS) imagery. This design captures fine-grained geospatial features, enabling reasoning across broader spatial regions, an aspect often overlooked in prior work.
> > >
> > > It is noteworthy that GeoPixel is not a minor extension of existing approaches *but rather a purpose-built solution for geographic reasoning* with a demonstrated impact on downstream tasks. RS often involves positional references (e.g., "the red car at top left"), requiring models to resolve spatial relationships among similar objects. GeoPixel demonstrates robust grounding in such cases both quantitatively and qualitatively. As demonstrated in Table 3 and Figure 9, tiling does not impair the model’s ability to localize objects based on positional language; in fact, it improves generalization.
> > >
> > > Furthermore, a key contribution of our work is the introduction of a *semi-automated annotation pipeline* specifically designed for remote sensing imagery. This pipeline leverages spatial priors and regional markers to generate high-quality pixel-level annotations aligned with natural language efficiently. As a result, we present the first pixel-grounded remote sensing dataset that supports multi-object segmentations interleaved with language, addressing a critical gap.
> > >
> > > **On Experiment and Ablation Study:**
> > >
> > > Geopixel, being a segmentation specialist model, is evaluated against a SOTA model across four complex tasks—RS-GCG, RES, Referring Expression Horizontal Detection, and Referring Expression Oriented Detection. In RS-GCG, it outperforms expert models (LISA, PixelLM, and GlaMM) under fair modification and training conditions evaluated across diverse metrics. Notably, in RES, GeoPixel surpasses specialized models, marking first among RS-focused LMMs and achieving a significant margin (*+15.83* Acc\@0.5 test) over concurrent work like GeoGround on the RRSIS-D benchmark.
> > >
> > > Geopixel outperforms recent LMMs (Geochat, GeoPix) in referring object detection (grounded localization through HBB) by leveraging superior pixel-level alignment. Moreover, in RS, oriented object detection is crucial, as aerial views often contain densely packed objects in arbitrary directions. Mainstream detectors use five- or eight-parameter formats, while current LMMs output quantized versions of these. We reformulate the task as a post-processing step on segmentation masks, achieving significant gains (*+33.7* Acc\@0.5) over the state-of-the-art.
> > >
> > > Regarding ablation studies on resolution and tiling, we clarify the interpretation of Table 4 and our training methodology. Following [1], the 'training patch number' specifies the maximum patch count permitted per sample during training. Critically, our training protocol employs patching at multiple scales: patches are extracted not only at the nominal maximum size but also from lower-resolution variants of the input data. This ensures the model is exposed to a spectrum of resolutions up to that defined by the maximum patch configuration. Consequently, we contend that the ablation study performed on inference-time patch size comprehensively evaluates the impact of effective input resolution and sufficiently demonstrates the utility of tiling within our framework.
> > >
> > > LMMs require effective modality alignment between visual and language tokens. In our model, this alignment is achieved through a vision projection layer and additionally through pLoRA [1]. To study the effect of this alignment for RS data, we conducted an ablation on the training of the vision projection layer (a 2-layer MLP), with results presented in Table 6. These results highlight the role of feature alignment. It is crucial to note that the pLoRA, pre-trained on extensive data in [1], was maintained without modification, thus serving as a consistent auxiliary alignment factor.
> > >
> > > [1] https://arxiv.org/abs/2401.16420

---

### Official Review · Reviewer_Amru · 2025-03-16

**Overall Recommendation:** 3

**Summary:**

The paper presents GeoPixel, a novel large multimodal model (LMM) that advances high-resolution remote sensing (RS) image analysis by integrating pixel-level grounding with textual understanding, addressing limitations in existing RS-LMMs. Its architecture features an adaptive image divider for processing 4K-resolution RS images through global-local patch fusion, combined with vision encoders (CLIP ViT-L and SAM-2), the InternLM2 language model, and a pixel decoder for mask generation. GeoPixel demonstrates excellent performance on RS-GCID tasks and RRSIS segmentation tasks; additionally, a key contribution is the semi-automatically constructed dataset GeoPixelD, which serves as an effective resource for pixel-level understanding of RS imagery.

**Claims And Evidence:**

Claim: GeoPixel supports high-resolution RS imagery.
Evidence: Dynamic partitioning (up to 4K) and experiments showing performance gains with higher patch counts (Table 4).

Claim: Pixel-level grounding improves RS understanding.
Evidence: Comparisons with bounding-box-based RS-LMMs (Table 1) and superior metrics in multi-target segmentation (Table 2).

Claim: GeoPixelD enables fine-grained comprehension.
Evidence: Dataset construction details (Section 4) and ablation on annotation complexity (Table 5).

**Essential References Not Discussed:**

There are some MLLM with Segmentation Capability works [1][2][3].
[1] RSUniVLM: A Unified Vision Language Model for Remote Sensing via Granularity-Oriented Mixture of Experts.
[2] GeoGround: A Unified Large Vision-Language Model. for Remote Sensing Visual Grounding.
[3] GeoPix: Multi-Modal Large Language Model for Pixel-level Image Understanding in Remote Sensing.
Discussing the differences in tasks and methods can help readers better understand this article.

**Experimental Designs Or Analyses:**

Yes, Baselines (LISA, PixelLM) are adapted for RS via fine-tuning on GeoPixelD, ensuring fair comparison. The RRSIS benchmark (Table 3) uses established metrics (P@0.5, mIoU), but cross-dataset generalization (e.g., on non-iSAID data) is untested.

**Methods And Evaluation Criteria:**

Methods: Adaptive partitioning, dual vision encoders, and a pixel decoder address RS-specific challenges, constructing a pipeline to achieve image understanding and pixel-level segmentation tasks.

Evaluation: Metrics (CIDEr, mIoU) and benchmarks (RS-GCG, RRSIS) align with task objectives. Baselines (LISA+, PixelLM+) are adapted fairly but require clarity regarding their modifications.

**Other Comments Or Suggestions:**

Please see Weaknesses. The experimental results of the article are good, the workload is sufficient, and the quality of the data set is also high. My focus may be on the effectiveness and significance of the task itself.

**Other Strengths And Weaknesses:**

Strengths:
1. First RS-LMM to achieve pixel-level grounding, addressing a critical gap in existing literature.
2. GeoPixelD’s semi-automated annotation pipeline ensures high-quality, granular data for RS-specific tasks.
3. The paper is well-presented and well-written.
4. It is good that the author considered the limitations and challenges

Weaknesses:
1. It is unclear what GeoPixel has done with remote sensing images and how effective they are.
2. There is a lack of discussion with some related work.(Please see Essential References Not Discussed *)
3. No discussion of training/inference efficiency, especially for the additional SAM2 decoder.

**Questions For Authors:**

1. The paper mentions adaptive partitioning, but the multi-scale feature fusion method is widely used in MLLM, such as S2-Wrapper[1]. Is there any special design for remote sensing? Or is this design more helpful for remote sensing images?
[1] Shi B, Wu Z, Mao M, et al. When do we not need larger vision models?

2. Do we need GCG tasks for remote sensing images? The current mainstream supervised segmentation tasks often target limited categories in remote sensing images. In comparison, what are the advantages of GEOPIXEL, which is also oriented to a single task?

3. Experiments rely heavily on GeoPixelD/iSAID; cross-dataset robustness (e.g., DIOR, xView) is untested. Can GeoPixel be extended to more general remote sensing scenarios?

**Relation To Broader Scientific Literature:**

GeoPixel advances RS-LMMs by introducing pixel grounding, addressing a gap in prior works (e.g., RSGPT, GeoChat) limited to bounding boxes. It aligns with natural-image pixel-grounding models (LISA, PixelLM) but is applied to remote sensing scenarios.

**Theoretical Claims:**

The paper does not make any theoretical claims that require rigorous mathematical proof. Its core contributions focus on method design and empirical verification.

---

> ### Author Rebuttal · Authors · 2025-04-01
>
> Dear Reviewer Amru,
>
> Thank you for your detailed review and constructive comments regarding our submission on GeoPixel. We appreciate your taking the time to analyze our work and the insights you provided. Below, we address your comments and concerns:
>
> **Clarity on Adapted Baselines:** In our experiments, LISA+ extends LISA which couldn't handle multiple instances and was enhanced to include multitarget segmentation masks in its output pipeline and phrase tokens (<p> and </p>) are added for the GCG task.  PixelLM+ builds on PixelLM, where phrase tokens are added, and <SEG> token is replaced with multiple codebook tokens. These changes ensure a fair comparison and are detailed in Section 5.2 (Baselines) of our manuscript.
>
> **Related Work:** GeoGround, RSUniVLM, and GeoPix (which was published in January 2025) are works concurrent to ours and share similarities. Comparative analysis can enrich the understanding of GeoPixel's unique contributions. GeoGround and RSUniVLM support pixel-level grounding by converting masks into text sequences, adding a computational burden to the LLM that scales with the number of distinguishable objects. GeoPixel resolves this limitation through end-to-end training with a dedicated mask decoder. Moreover, GeoPixel not only outperforms GeoGround on RRSIS-D data **+11.11**(Acc0.5 val), **+15.83**(Acc0.5 test), **+6.89**(mIoU val), and **+6.8**(mIoU test) but also outperforms specialist models (Table 4) showcasing strong referring expression segmentation capability.
>
> **Training and Inference Efficiency:** All models, including GeoPixel and the baseline models (LISA+, PixelLM+, and GLaMM-ft), were trained for the same epochs with similar computational resources, ensuring performance gains stem from model design, not training or hardware differences. Furthermore, to provide a comparison of inference efficiency, we additionally report average runtime per sample: LISA+: 25.28s, PixelLM+: 91.44s, GLaMM-ft:35.08s, and GeoPixel: (P=1) 46.48s for RS-GCG task.
>
> **Adaptive Partitioning and Multi-scale feature fusion MSFF:** Adaptive partitioning is not merely an alternative but a complementary strategy to MSFF. RS imagery exhibits significant variations in spatial coverage (e.g. 800x800 pixels can cover diverse areas such as 1 km² or 10 km²). In this context, image resolution is crucial to determine the granularity and clarity of details within each square meter. High resolution (HR) enhances the visibility of finer features, essential for accurate identification and detailed analysis. MSFF can also be valuable by allowing information integration from various scales. Balancing both strategies presents a promising direction for future exploration, potentially improving RS accuracy and efficiency.
>
> **Necessity and Advantages of GCG in RS:** GCG is immensely important in RS, as traditional supervised segmentation tasks, although precise, are limited to fixed categories and offer minimal interactivity. GCG adds an intuitive, interactive layer, enabling users to explore and query data more flexibly, broadening the scope beyond predefined labels. GeoPixel demonstrates specialized, high-precision segmentation capability, but by integrating GCG, it can expand its usability, empowering more dynamic and user-driven data exploration.
>
> **Cross-Dataset Robustness:** We evaluated GeoPixel on VRSBench referring expression detection task, reporting Acc\@0.5 / Acc\@0.7 across 3 categories: unique (U), non-unique(NU) and overall (O)
>
> **MiniGPT-v2** scored 40.7/18.9 (U), 32.4/15.2 (NU), 35.8/16.8 (O).
>
> **LLaVA-1.5** achieved 51.1/16.4 (U), 34.8/11.5 (NU), 41.6/13.6 (O).
>
> **Mini-Gemini** showed lower performance with 41.1/9.6 (U), 22.3/4.9 (NU), 30.1/6.8 (O).
>
> **GeoChat** reached 57.4/22.6 (U), 44.5/18.0 (NU), 49.8/19.9 (O).
>
> **GeoPix** performed similarly with 57.0/22.7 (U), 44.8/18.2 (NU), 49.8/20.0 (O).
>
> **GeoPixel** achieved the best results: **70.37/41.54** (U), **65.80/40.32** (NU), **67.70/40.83** (O).
>
> We further compared **GeoChat** and **GeoPixel** on VRSBench using orientated bounding boxes.
>
> **GeoChat** scored 32.3/12.6 (U), 18.5/5.7 (NU), 24.3/8.6 (O).
>
> **GeoPixel** significantly outperformed, achieving **54.48/24.87** (U), **60.51/30.97** (NU), **58.00/28.42** (O).
>
> To avoid data leakage, GeoPixel (trained on GeoPixelD) is finetuned on VRSBench without using any data from RRSIS-D (a DIOR-based dataset). GeoPixelD and VRSBench rely on DOTA's training set for training and validation set for testing. These steps ensure no leakage from either DOTA or DIOR. We plan to extend our evaluations using additional datasets like xView to test GeoPixel's applicability to broader RS scenarios. The DIOR base results indicate promising adaptability, which we'll detail along with qualitative results in the revised submission.
>
> We hope our responses adequately address your concerns. Thank you for your recommendations, which have undoubtedly helped improve our work.

---

### Decision · Program_Chairs · 2025-05-01

**Decision:**

Accept (poster)

**Comment:**

After the discussion phase, the majority of reviewers recommended acceptance (Accept, 2x Weak Accept, Weak Reject), finding the paper to be well written, the introduced dataset valuable, and the proposed method to have novelty in its formulation. The remaining negative rating was due to concerns about the technical novelty (i.e., similarity to an existing method) and the lack of ablation studies to better highlight the contributions. The rebuttal addressed many of the reviewers' concerns, including those related to dataset annotation details.  Ultimately, the AC found the review by R-Amru to be the most compelling. This paper falls under the applications track, tackles an interesting problem, proposes a method with several novel elements, introduces a new dataset, and has compelling results. As such, an accept decision was reached. The AC does agree, however, that the ablations could be more robust. Please take the reviewer feedback into account when preparing the camera-ready version.